# Propolis alleviates ulcerative colitis injury by inhibiting the protein kinase C - transient receptor potential cation channel subfamily V member 1 - calcitonin gene-related peptide/substance P (PKC-TRPV1-CGRP/SP) signaling axis

Zhen Qian[1☯], Mengjie Zhang[2☯], Taiyu Lu[1], Jiayi Yu[1], Siyuan Yin[3], Haihua Wang[4], Jing Wang[◍][4]*

1 School of Clinical Medicine, Wannan Medical College, Wuhu, Anhui province, China, 2 Graduate School, Wannan Medical College, Wuhu, Anhui province, China, 3 School of Medical Imageology, Wannan Medical College, Wuhu, Anhui province, China, 4 Department of Physiology, School of Basic Medical Sciences, Wannan Medical College, Wuhu, Anhui province, China

☯ These authors contributed equally to this work.
* 247948083@qq.com

## Abstract

This study investigated the protective effect of water-soluble propolis (WSP) on colonic tissues in ulcerative colitis (UC) and the role of the protein kinase C - transient receptor potential cation channel subfamily V member 1 - calcitonin gene-related peptide/substance P (PKC-TRPV1-CGRP/SP) signaling pathway. Male SD rats were divided into a control group, a UC model group, various WSP groups (Low-WSP, Medium-WSP, and High-WSP) with UC, and a salazosulfapyridine (SASP) positive control group with UC. After UC was established, the WSP and SASP groups were treated with WSP or SASP, respectively, for 7 d. Each day, body weight measurements were obtained, and the disease activity index (DAI) was recorded by observing fecal characteristics and blood in the stool. After the experiment, hematoxylin and eosin (HE) colonic tissue staining was performed to observe pathological changes, western blotting and immunohistochemistry were performed to detect PKC, TRPV1, CGRP, and SP expression in colonic tissues, and laser confocal microscopy was performed to observe the fluorescence colocalization of PKC/TRPV1, TRPV1/CGRP, and TRPV1/SP. HE staining showed significant colonic tissue structure disruption and inflammatory infiltration in the UC group. Western blotting and immunohistochemistry showed that the expression of PKC, TRPV1, CGRP, and SP in the colonic tissues of the UC group increased significantly compared with that of the control group. Compared with the UC group, the expression of PKC, TRPV1, CGRP, and SP in colonic tissues was significantly reduced in the High-WSP, Medium-WSP, and SASP groups. Immunofluorescence showed the colocalized expression of PKC/TRPV1, TRPV1/CGRP, and TRPV1/SP proteins in the colon tissue of the UC group was significantly reduced after WSP and SASP interventions compared with that of the control group. The results suggest that the mechanism of

**Data Availability Statement:** All relevant data are within the paper and its Supporting Information files.

**Funding:** This work was supported by the Natural Science Research Project of Anhui Province (KJ2020A0606, KJ2016A729); College Students Innovation and Entrepreneurship Program of China (202110368034, 202310368023); College Students Innovation and Entrepreneurship Program of Anhui Province (S202210368135, S202310368001). The funders had no role in study design, data collection and analysis, decision to publish, or preparation of the manuscript.

**Competing interests:** The authors have declared that no competing interests exist.

UC alleviation by propolis may inhibit the PKC-TRPV1-CGRP/SP signaling pathway and the release of inflammatory mediators, thus alleviating inflammation.

## Introduction

UC is a chronic gastrointestinal dysfunctional disease with clinical manifestations of diarrhea, abdominal pain, mucopurulent stools, and decreased body mass; pathological manifestations of colonic UC include continuous, superficial, and diffuse characteristics. UC can easily recur and has a long disease course, which seriously affects the quality of life of patients [1,2]. The pathogenesis of UC may involve the interaction of environmental, genetic, infectious, intestinal micro-ecological, and immune factors, and the pathogenesis is not yet clear [3,4]. The clinical treatment of UC has no specific treatment plan, but it is common to protect the intestinal mucosa, control inflammatory effects, and remove pathogenic sources. However, the efficacy of treatment is often poor, and there are many side effects of treatment. The incidence of UC is increasing each year. Therefore, the search for safe and effective drugs for the treatment of UC is urgently needed.

Chemically-induced UC is commonly used,such as sodium dextran sulfate (DSS) -induced colitis, Acetic acid-induced colitis, oxazolidone-induced colitis, and 2,4,6 trinitrobenzene sulfonic acid (TNBS)/ ethanol-induced colitis. Those chemically-induced are all established and highly validated methods for UC model generation [5]. DSS induced ulcerative colitis was adopted in this experiment.

Propolis is a valuable resinous product that is produced by bees. Propolis contains hundreds of compounds including flavonoids, phenols, polyphenols, terpenoids, vitamins, and important minerals. These substances exhibit many biological activities [6]. Numerous preclinical and clinical studies have demonstrated that propolis has potential as an anti-cancer, anti-apoptotic, anti-diabetic, anti-inflammatory, antioxidant, antibacterial, and antiviral agent [7].

TRPV1 is a ligand-gated nonselective cation channel widely present in the cell and organelle membranes of the central and peripheral nervous systems. TRPV1 can be activated by injurious stimuli and is one of the main mediators of inflammation-induced pain [8,9]. Activation of TRPV1 channels triggers a large inward flow of calcium ions, causing action potentials at nerve endings and leading to the release of the injurious inflammatory transmitters CGRP and SP. TRPV1 is involved in several digestive disorders, including inflammatory gastrointestinal diseases [9,10].

PKC is a class of cytoplasmic serine/threonine kinases that are normally present in the cytoplasm in an inactive form. PKC is activated by phosphorylation of serine/threonine of a variety of proteins and is involved in cellular messaging, ion channel regulation, cell growth and differentiation, metabolism, and transcriptional activation, thereby affecting intracellular biometabolism. PKC is a key regulator of activation and desensitization of TRPV1 [11]. TRPV1 activation is dependent on protease-activated receptor 2, which leads to increased visceral sensitivity [12,13] and nociceptive hypersensitivity [14,15].

CGRP and SP are important neuroendocrine peptides of the brain-gut axis. CGRP and SP are involved in esophageal visceral hypersensitivity through an inflammatory response. Overexpression of CGRP and SP in esophageal mucosal peripheral nerves after stimulation by chemicals or mechanical dilation is accompanied by a large accumulation of inflammatory factors, causing a local inflammatory response [16,17].

Our group found that WSP has a good protective effect against colonic injury in UC, and its protective mechanism is related to the downregulation of TRPV1 expression and the inhibition of inflammatory mediator release. In this study, we further explored the upstream molecules of TRPV1, PKC, and the downstream molecules, CGRP and SP, to determine whether the protective mechanism of propolis against UC is related to the PKC-TRPV1-CGRP/SP signaling pathway.

## Materials and methods

### Animal grouping and UC modeling

This study was approved by the Animal Ethics Committee of Wanan Medical College (LLSC-2021-024) and performed in strict accordance with the guidelines of the Animal Care and Use Committee of Wanan Medical College. Forty-eight male SD rats (SPF grade, 200 ± 20 g) purchased from Hangzhou Medical College, (Certificate of Conformity: SCXK (Zhe) 2019–0002), were fed ad libitum with diet and water before the experiment and acclimatized for 1 week. The rats were then randomly divided into six groups (n = 8): (1) Normal control group (NC group): free drinking water during the experiment. (2) UC model group (UC group): The rats were fed 4% dextran sulfate sodium (DSS) for 7 d and continued to drink water ad libitum for 7 d. (3) Low-WSP group (L-WSP group): Based on the successful replication of the UC group, 50 mg/kg of WSP was administered each day via gavage for 7 d. (4) Middle-WSP group (M-WSP group):100 mg/kg of WSP was administered each day by gavage for 7 d based on successful replication of the UC group. (5) High-WSP group (H-WSP group): Based on the successful replication of the UC group, 200 mg/kg of WSP was administered each day by gavage for 7 d. (6) SASP positive control group (SASP group):100 mg/kg of SASP solution was administered by gavage each for 7 d based on successful replication of the UC group. After the last administration, all rats received fasting and free access to water for 24 hours. In order to alleviate the suffering of the animals, all animals were sacrificed by decapitation after receiving ethyl carbamate (1.5g/kg). An overview of the animal experiments is shown in Fig 1.

### Reagents

WSP (Item No. 20210325-B) was purchased from Guangzhou Jiehe Bee Co. Ltd. (Guangzhou, China) and SASP (Cat. No. 09210103) was purchased from Shanghai Xinyi Tianping Pharmaceutical Co. (Shanghai, China). Primary antibodies Anti-PKC (Cat. No. ab184746) and Anti-TRPV1 (Cat. No. ab203103) were purchased from Abcam(England); the primary antibodies Anti-CGRP (Cat. No. PA5-114929), and anti-SP (Cat. No. PA5-106934) were purchased from Invitrogen(USA), and horseradish peroxidase (HRP)-cross-linked IgG secondary antibodies (Cat. No. BL001A, BL003A), and immunohistochemistry kits (Cat. No. BL729A, BL733A) were purchased from Biosharp (China). Immunofluorescent secondary antibodies 594-conjugated AffiniPure Donkey Anti-Mouse (Cat. No. 715-585-150) and 488-conjugated AffiniPure Donkey Anti-Rabbit (Cat. No. 711-545-152) were purchased from the Jackson Corporation (USA).

### Observation of body weight and stool properties of rats

Rats were observed daily at the beginning of the experiment for water intake, diet, and stool properties. The body weights of the animals were measured at the same time point every day, and the weight on day 1 was marked as the pre-experimental starting weight. The percentage change in the body weight of rats in each group was calculated using the following formula:

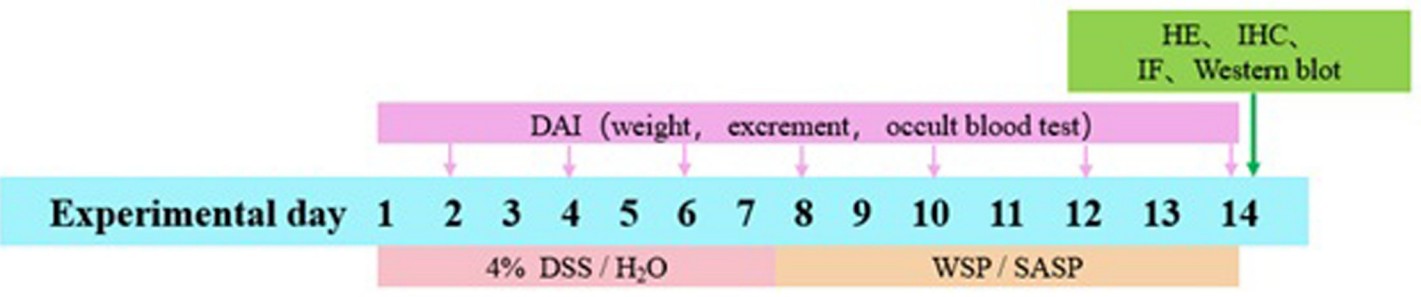

**Fig 1. Schematic diagram of animal experiment.**

Body weight change (%) = [(body weight on a day after the experiment - starting weight before the experiment)/starting weight before the experiment] × 100%.

### Stool occult blood test

Blood in rat stools was detected using an occult blood test kit. Briefly, rat stool was applied to a filter paper, and after adding drops of mixture A and B of the kit, the color development results were determined according to the following criteria: (1) dark violet-blue immediately after adding the reagent (++++); (2) violet-blue (+++) within 10 s after adding the reagent; (3) violet-red (++) within 1 min after adding the reagent; (4) purple-red color (+) within 1–2 min after adding the reagent; (5) no color after 2 min of adding the reagent (–).

### DAI score

Based on the descriptions in the literature [18], DAI was assessed according to the following criteria based on weight change, stool properties, and the amount of blood in the stool from day 2: Score 0: Weight loss ≤ 1%, normal stool, blood in stool (-); Score 1: Weight loss 1–5%, soft but formed stool, blood in stool (+); Score 2: Weight loss 5–10%, loose stool, blood in stool (++); Score 3: Weight loss 10–15%, very loose stool, blood in stool (+++); Score 4: Weight loss > 15%, watery stool, blood in stool (++++).

### Specimen collection

After the last administration, the animals were fasted without water for 24 h and then sacrificed, and colonic tissues were collected. The colonic tissues were cleaned with saline at 4°C, and some of the colonic tissues were stored at -80°C and used for western blot experiments. A portion of the tissue was fixed in 4% paraformaldehyde for HE staining, immunohistochemistry, and immunofluorescence experiments. Four percent paraformaldehyde-fixed colon tissues were dehydrated in 15% and 30% sucrose solutions for one day each, embedded in OCT, cut into 8 μm sections using a Leica frozen section machine(Germany), and stored at -20°C.

### HE staining

The frozen sections were stored at -20°C and dried at room temperature, fixed by PBST for 10 min, stained with hematoxylin for 6 min, washed with water, fractionated by hydrochloric acid for 10 s, stained with eosin for 20 s, washed with water, dehydrated using an alcohol gradient (70%→75%→80%→85%→90%→100%) for 1 min each, made transparent with xylene, sealed, and the structural changes of colon tissue were observed using a Leica DMi8-THUN-DER imager (Germany.

## Western blot

One microliter of lysate was added to 0.1 g of colon tissue, and the tissue was homogenized and left for 30 min at room temperature. The supernatant was collected by centrifugation at 12000 rpm for 30 min at 4˚C. Ten micrograms of total protein was separated by SDS-PAGE, transferred to a PVDF membrane, blocked with skim milk, incubated with primary antibody overnight at 4˚C, incubated with secondary antibody for 1 h after three washes with TBST, exposed to an ECL developer, photographed, and analyzed by ImageJ 5.0 software for protein band grayscale values.

## Immunohistochemistry

Immunohistochemical methods are available everywhere [19]. In brief, sections stored at -20˚C were dried at room temperature and washed three times with PBST for 10 min each time. The sections were treated with peroxidase inactivator (30% $H_2O_2$: methanol = 1:9) for 30 min and then washed with PBST three times for 10 min each time. The sections were subjected to antigen repair with citrate buffer and washed with PBST three times for 10 min each time. Sections were closed with 0.5% BSA solution for 30 min, incubated overnight at 4˚C by adding the corresponding primary antibody, and washed with PBST three times for 10 min each time while shaking. The sections were then incubated with HRP-Polymer for 20 min at 37˚C and washed with PBST three times for 2 min each time. Sections were developed with DAB chromogen for 10 min followed by hematoxylin staining for 10 s, washing with water, alcohol gradient dehydration (70% → 80% → 90%) for 1 min each, 100% alcohol dehydration two times for 3 min each, and made transparent with xylene. After the sections were encapsulated in neutral resin, the expression of PKC, TRPV1, CGRP, and SP in the colon was observed using a Leica DMi8-THUNDER imager(Germany). The relative expression of the target proteins was analyzed using the Image Pro Plus 6.0 software.

## Immunofluorescence colocalization

Immunohistochemical methods were employed for the detection of the targets according to the methods in the literature [20]. Sections were baked at 37˚C for 2 h and then treated with 3% H2O2 for 10 min to inactivate peroxidase. Sections were shaken and washed with PBST three times for 5 min each and then bathed in citrate buffer in a 95˚C water bath for 8 min for antigen repair. After washing with PBST three times for 5 min each, sections were blocked with horse serum for 30 min and incubated overnight at 4˚C with the corresponding primary antibody. The sections were then washed three times with PBS for 5 min each and incubated with the HPR-cross-linked IgG secondary antibody at room temperature for 2 h. The fluorescent expression of PKC/TRPV1, TRPV1/CGRP, and TRPV1/SP was observed using a SP8 laser confocal microscope(Germany). Fluorescence intensity was analyzed for colocalized expression using the Image Pro Plus 6.0 software.

## Data analysis

Data were expressed as mean ± standard deviation, and SPSS18.0 statistical software was used to analyze the data. One-way ANOVA and repeated-measures ANOVA were used for comparisons between multiple groups, and the LSD test was used for two-way comparisons between multiple groups. $p < 0.05$ was considered a statistically significant difference.

# Results

## DAI scores in UC rats

The DAI scores of rats in each group are shown in Fig 2. Compared to the normal control group, differences in DAI scores were observed in the remaining groups of rats after 2 d of free DSS consumption (p < 0.05). Four days later, highly significant differences in the DAI scores were observed in the remaining groups (p < 0.001). Compared with the model group, differences were observed in the H-WSP and SASP groups at 3 d of intervention (p < 0.05), and highly significant differences were observed in the H-WSP, M-WSP, and SASP groups at 5 d of intervention (p < 0.001).

## Effects of propolis on the structure of colonic tissues

The effects of propolis on colonic tissue structures are shown in Fig 3. In the NC group, the colonic tissue of rats with neatly arranged glands and a clear crypt structure showed no inflammatory infiltration. In the UC group, the colonic tissues showed obvious inflammation and ulceration, partial disappearance of glands, severe inflammatory infiltration, epithelial cell shedding, and severe damage to the crypt. The WSP group showed a small amount of epithelial detachment and scattered inflammatory infiltrates.

## Expression of TRPV1, PKC, CGRP, and SP protein in colonic tissues

The effects of propolis on TRPV1, PKC, CGRP, and SP expression in UC-derived colonic tissues are shown in Fig 4–1. Compared to the NC group, high expression of TRPV1, PKC, CGRP, and SP was observed in the colonic tissues of rats in the UC group (p < 0.001). Compared with the UC group, both WSP and SASP interventions inhibited the expression of four proteins, TRPV1, PKC, CGRP, and SP (p < 0.001, p < 0.01, respectively) (Fig 4–2).

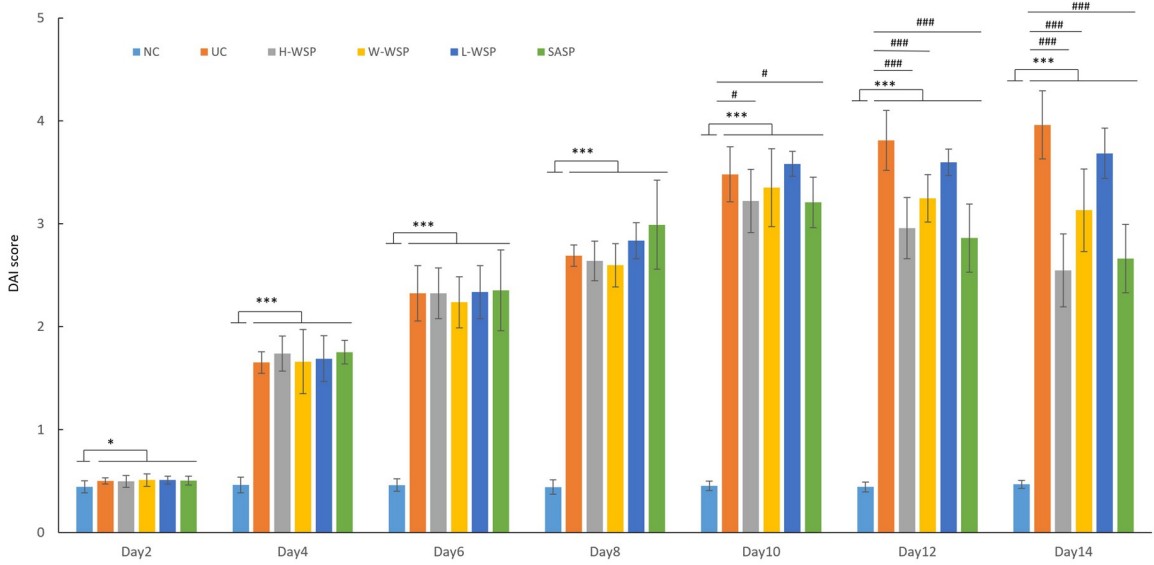

**Fig 2. The DAI score of rats (n = 8).** NC: Normal control group;UC: Ulcerative colitis model group;H-WSP: High-WSP group;M-WSP: Middle-WSP group;L-WSP: Low-WSP group, SASP: Salazosulfapyridine control group *Compared with NC group; #Compared with UC group. $^*P < 0.05$, $^{***}P < 0.001$ *vs* NC group; $^\#P < 0.05$, $^{\#\#\#}P < 0.001$ *vs* UC group.

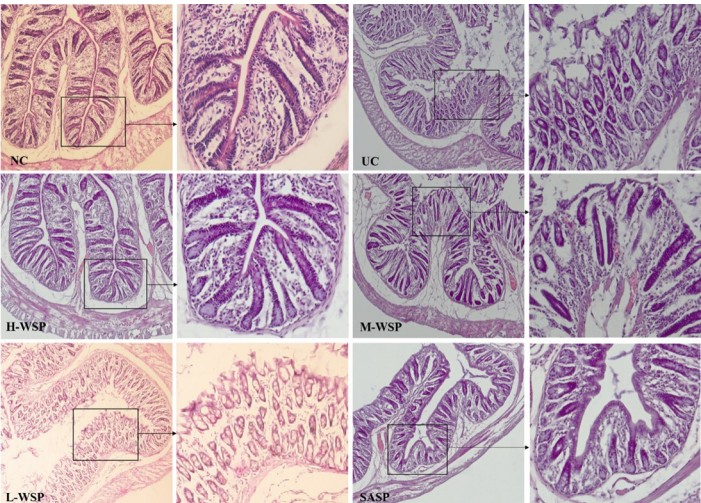

**Fig 3. Effect of propolis on colonic tissue structure.** NC: Normal control group;UC: Ulcerative colitis model group; H-WSP: High-WSP group;M-WSP: Middle-WSP group;L-WSP: Low-WSP group, SASP: Salazosulfapyridine control group. The glands were aligned without inflammatory infiltration in NC group. The glands were partially absent, severely damaged, and had severe inflammatory infiltration in UC group. The structure of the glands were basically restored and aligned in H-WSP and SASP groups.

## Immunohistochemical Expression of TRPV1, PKC, CGRP, and SP protein in colonic tissues

Immunohistochemical results are shown in Fig 5–1. Compared to the NC group, PKC, TRPV1, CGRP, and SP were highly expressed in the colon tissue of the UC group ($p < 0.001$). In contrast, WSP and SASP inhibited the expression of PKC, TRPV1, CGRP, and SP compared to those in the UC group ($p < 0.05$ and $p < 0.001$ respectively). (Fig 5–2).

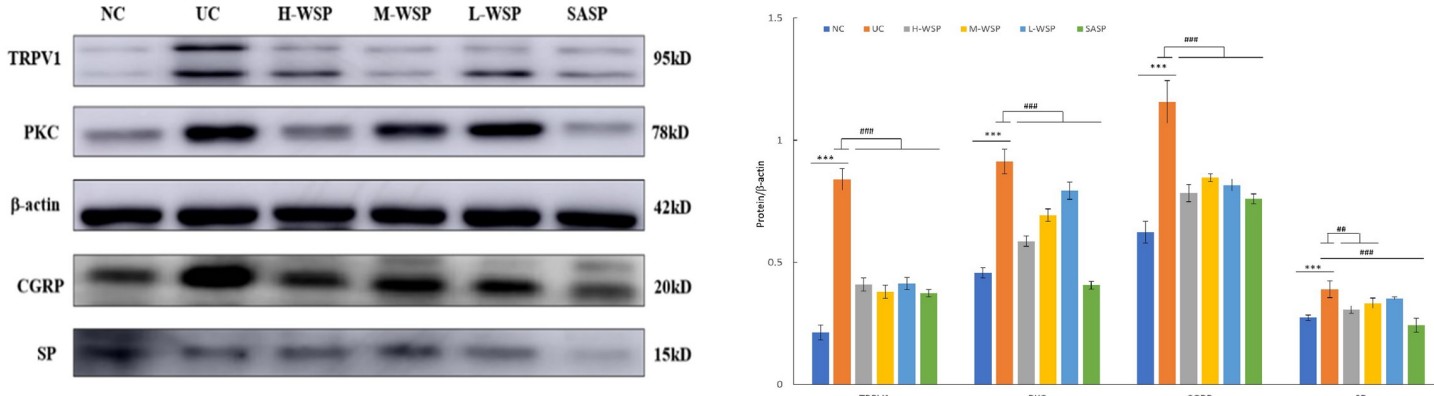

**Fig 4. Expression of TRPV1, PKC, CGRP, and SP in UC-derived colonic tissues (n = 3).** (1) The western blot results of TRPV1, PKC, CGRP, and SP in colon tissue (2) Differences in protein expression among groups NC: Normal control group;UC: Ulcerative colitis model group;H-WSP: High-WSP group;M-WSP: Middle-WSP group; L-WSP: Low-WSP group, SASP: Salazosulfapyridine control group *Compared with NC group; #Compared with UC group ***$P<0.001$ *vs* NC group; ##$P < 0.01$,###$P<0.001$ *vs* UC group.

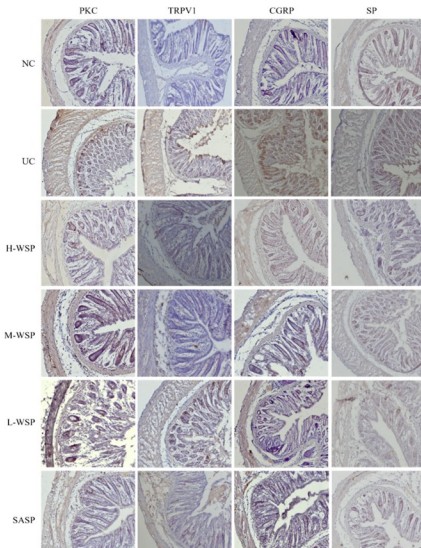
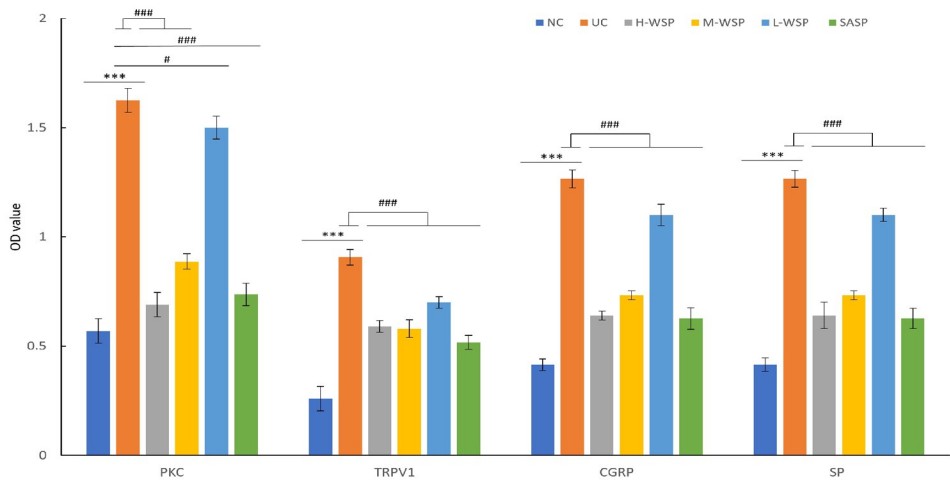

**Fig 5. Immunohistochemical Expression of TRPV1, PKC, CGRP, and SP protein in colonic tissues(n = 3).** (1) The immunohistochemical results of TRPV1, PKC, CGRP, and SP in colon tissue (2) Differences in protein expression among groups. NC: Normal control group;UC: Ulcerative colitis model group;H-WSP: High-WSP group;M-WSP: Middle-WSP group;L-WSP: Low-WSP group, SASP: Salazosulfapyridine control group. *Compared with NC group; #Compared with UC group
***$P<0.001$ *vs* NC group; #$P<0.05$, ###$P<0.001$ *vs* UC group.

## Colocalization of PKC/TRPV1, TRPV1/CGRP, TRPV1/SP in colonic tissues

The colocalization results are shown in Figs 6–9. Compared with the NC group, the degree of protein colocalization of PKC/TRPV1, TRPV1/CGRP, and TRPV1/SP in the colonic tissue in the UC group was significantly increased (p < 0.001). After WSP and SASP interventions, the protein colocalization expression of the three groups was significantly reduced (p < 0.001,

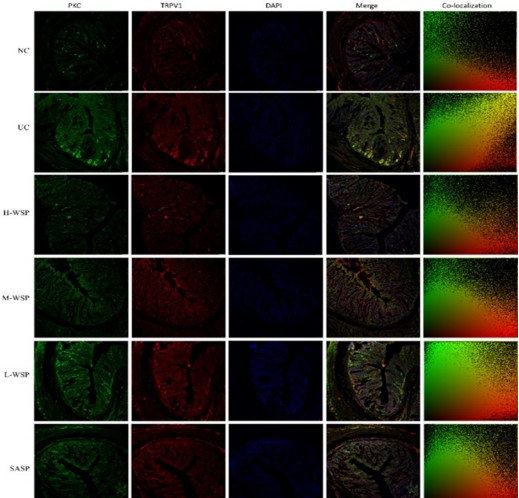

**Fig 6. Colocalization of PKC/TRPV1 in colonic tissue (n = 3).** NC: Normal control group;UC: Ulcerative colitis model group;H-WSP: High-WSP group;M-WSP: Middle-WSP group;L-WSP: Low-WSP group, SASP: Salazosulfapyridine control group.

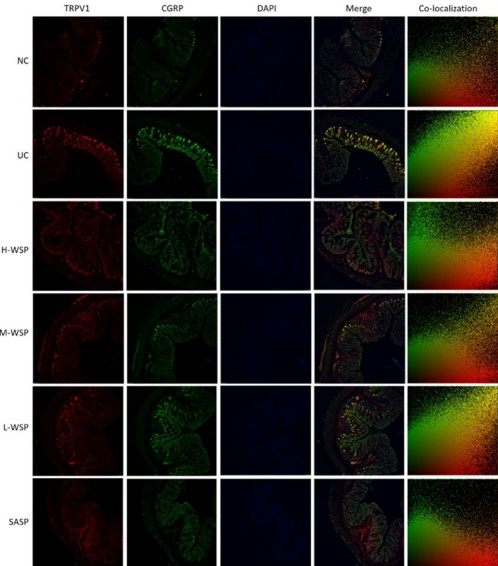

**Fig 7. Colocalization of TRPV1/CGRP in colonic tissue (n = 3).** NC: Normal control group;UC: Ulcerative colitis model group;H-WSP: High-WSP group;M-WSP: Middle-WSP group;L-WSP: Low-WSP group, SASP: Salazosulfapyridine control group.

$p < 0.01$, and $p < 0.05$, respectively), and the H-WSP and SASP groups showed highly significant differences ($p < 0.001$).

## Discussion

The pathogenesis of UC is a multi-factor process, including genetic factors, environmental factors, diet and living habits etc. Intestinal microbial ecological disorders, immune homeostasis

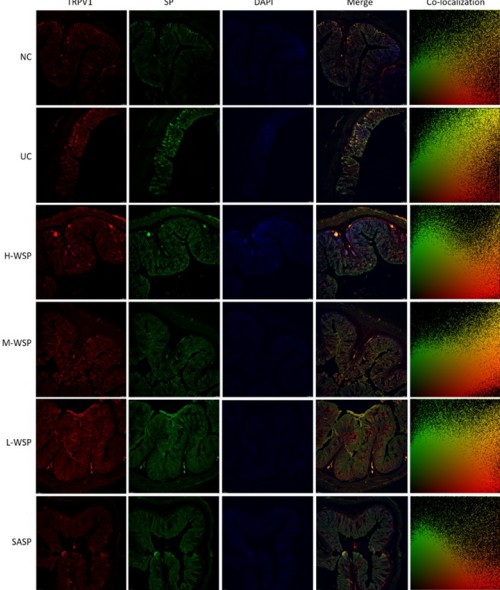

**Fig 8. Colocalization of TRPV1/SP in colonic tissue (n = 3).** NC: Normal control group;UC: Ulcerative colitis model group;H-WSP: High-WSP group;M-WSP: Middle-WSP group;L-WSP: Low-WSP group, SASP: Salazosulfapyridine control group.

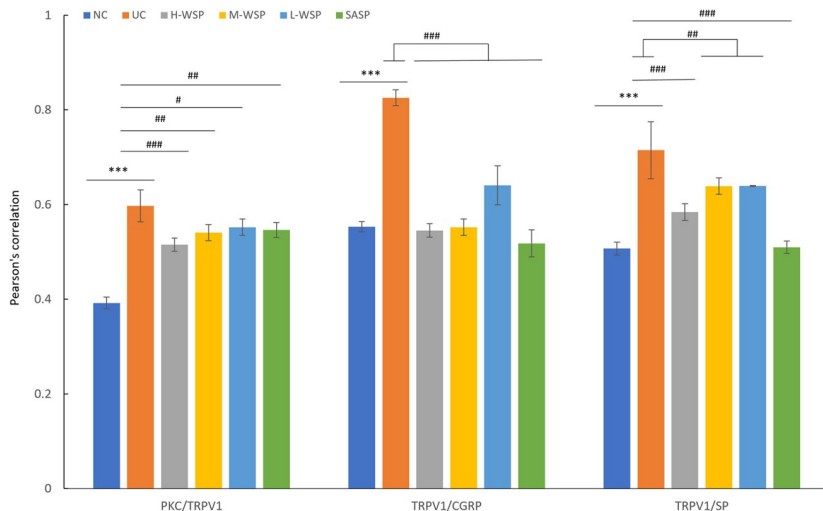

**Fig 9. Quantification of colocalization of PKC/TRPV1, TRPV1/CGRP, TRPV1/SP.** NC: Normal control group;UC: Ulcerative colitis model group;H-WSP: High-WSP group;M-WSP: Middle-WSP group;L-WSP: Low-WSP group, SASP: Salazosulfapyridine control group *Compared with NC group; #Compared with UC group ***$P<0.001$ *vs* NC group; #$P<0.05$,##$P<0.01$,###$P<0.001$ *vs* UC group.

disorders, and intestinal mucosal barrier changes play key roles in the pathogenesis of UC, and the adjustment of intestinal flora and immune homeostasis can improve UC symptoms [21,22]. Propolis has been shown to have a protective effect in animal models of UC, as evidenced by a reduction in inflammatory infiltration, histological damage, expression of inflammatory mediators, inflammatory enzyme activity, and a decrease in inflammatory markers [23,24]. Similar results were observed in this study. It is clear from the DAI score results that propolis is also effective in ameliorating inflammation in a UC mouse model.

Capsaicin-sensitive sensory neurons contain peptide transmitters such as SP and CGRP [25]. TRPV1 receptors (also called capsaicin receptors) can be activated by toxic heat, protons, and vanilloids (capsaicin, resinin). The subsequent activation of TRPV1 channels induces the release of neuropeptide mediators, allowing capsaicin-sensitive nerves to perform local efferent functions, leading to neurogenic inflammation [26,27]. Studies have confirmed that TRPV1 is highly expressed in rodent models of inflammatory bowel disease and UC [28,29]. Our data also showed that TRPV1 expression was elevated in UC colonic tissue. In addition, we found that peptide transmitters, such as SP and CGRP, were significantly elevated in the colonic tissues of UC animal models. Their expression significantly decreased after the administration of propolis.

The role of protein kinase Cδ in promoting inflammation has been widely demonstrated [29–31]. An in vivo study found that propolis downregulated protein kinase Cδ expression in an animal model of rhinitis [32]. Our results showed that PKC was highly expressed in colonic tissue in the UC model, and propolis administration significantly reduced PKC expression.

In conclusion, this study confirmed that the PKC-TRPV1-CGRP/SP signaling pathway appears to be highly expressed in colonic tissues with UC and is involved in the signaling of UC; therefore, we hypothesized that the mechanism of UC may be related to visceral hypersensitivity. The results also confirmed that the expression of the PKC-TRPV1-CGRP/SP signaling pathway was downregulated after WSP and SASP interventions; therefore, it is speculated that WSP may inhibit PKC, downregulate the expression of TRPV1, CGRP, and SP, and reduce the

release of injurious neurotransmitters by inhibiting the PKC-TRPV1-CGRP/SP signaling pathway, decreasing visceral sensitivity and alleviating pain and inflammatory ulcer damage in UC.

## Supporting information

**S1 File. Fig 2. DAI raw data.**
(DOCX)

**S2 File. Fig 5. Immunohistochemical raw data.**
(DOCX)

**S3 File. Fig 9. Colocalization raw data.**
(DOCX)

**S1 Raw images.**
(PDF)

## Author Contributions

**Data curation:** Zhen Qian, Mengjie Zhang, Taiyu Lu, Jiayi Yu, Siyuan Yin.

**Formal analysis:** Zhen Qian, Taiyu Lu, Jiayi Yu.

**Investigation:** Zhen Qian, Mengjie Zhang, Taiyu Lu, Jiayi Yu, Jing Wang.

**Methodology:** Mengjie Zhang.

**Project administration:** Haihua Wang.

**Resources:** Mengjie Zhang.

**Supervision:** Haihua Wang.

**Visualization:** Jing Wang.

**Writing – review & editing:** Jing Wang.

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
