## [Decision Letter · Decision Letter 0]

23 May 2023

PONE-D-23-03632Propolis alleviates ulcerative colitis injury by inhibiting the protein kinase C - transient receptor potential cation channel subfamily V member 1 - calcitonin gene-related peptide/substance P (PKC-TRPV1-CGRP/SP) signaling axisPLOS ONE

Dear Dr. Haihua,

Thank you for submitting your manuscript to PLOS ONE. After careful consideration, we feel that it has merit but does not fully meet PLOS ONE’s publication criteria as it currently stands. Therefore, we invite you to submit a revised version of the manuscript that addresses the points raised during the review process.

Please revise the manuscript and resubmit.==============================

We look forward to receiving your revised manuscript.

Kind regards,

Suprabhat Mukherjee, Ph.D.

Academic Editor

PLOS ONE

3. Please ensure that you include a title page within your main document. We do appreciate that you have a title page document uploaded as a separate file, however, as per our author guidelines (http://journals.plos.org/plosone/s/submission-guidelines#loc-title-page) we do require this to be part of the manuscript file itself and not uploaded separately.

“This work was supported by the Natural Science Research Project of Anhui Province(KJ2020A0606, KJ2016A729); College Students Innovation and Entrepreneurship Program of China（202110368034）; College Students Innovation and Entrepreneurship Program of Anhui Province (S201910368090, S201910368113).”

6. PLOS requires an ORCID iD for the corresponding author in Editorial Manager on papers submitted after December 6th, 2016. Please ensure that you have an ORCID iD and that it is validated in Editorial Manager. To do this, go to ‘Update my Information’ (in the upper left-hand corner of the main menu), and click on the Fetch/Validate link next to the ORCID field. This will take you to the ORCID site and allow you to create a new iD or authenticate a pre-existing iD in Editorial Manager. Please see the following video for instructions on linking an ORCID iD to your Editorial Manager account: https://www.youtube.com/watch?v=_xcclfuvtxQ.

Additional Editor Comments:

1. Introduction: It is poorly written. It should contain the following:

a. Pharmacology of Propolis.

b. Mention the typical aetiology of UC and link PKC-TRPV1-CGRP/SP signaling pathway to that.

c. Authors need to cite a recent literature (https://doi.org/10.1016/j.mex.2023.102158) regarding chemical induced experimental UC generation.

2. Methods: Please revise the subtitles. E.g. ‘Stool monitoring ‘ is not appropriate.

3. Results: Change the subtitle “DAI scores in UC mice”. “Effect of propolis on the expression of TRPV1, PKC, CGRP, and SP in UC-derived colonic tissues” should be changed to “Expression of ……………………………CGRP protein”. Immunofluorescence and immunohistochemistry need proper elaboration.

4. Discussion should include 2-3 lines on the Gut microbial dysbiosis as a cause of IBD and UC. UC also increases the chance of colorectal cancer. Author should follow and include the two following citations : https://doi.org/10.1016/j.jnutbio.2018.07.010 and https://doi.org/10.3390/vaccines11030525

5. Figure: Please improve the captions and the quality of the immunoblot.

Reviewers' comments:

Reviewer's Responses to Questions

**Comments to the Author**

1. Is the manuscript technically sound, and do the data support the conclusions?

Reviewer #1: Yes

2. Has the statistical analysis been performed appropriately and rigorously? 

Reviewer #1: Yes

3. Have the authors made all data underlying the findings in their manuscript fully available?

Reviewer #1: Yes

4. Is the manuscript presented in an intelligible fashion and written in standard English?

Reviewer #1: Yes

5. Review Comments to the Author

Reviewer #1: Dear Author

Thank you for your manuscript submission. Your work is well-designed and well-presented. Please do add all the related references in association with applied protocols in your manuscript. Minor Revision is needed.

6. PLOS authors have the option to publish the peer review history of their article (what does this mean?). If published, this will include your full peer review and any attached files.

Reviewer #1: **Yes: **Payam BEHZADI

---

## [Author Response · Author response to Decision Letter 0]

10 Jul 2023

Dear editor,

Thanks for your opinions on the revision of the paper, the revised situation is described as follows:

1. I have modified it as requested. 

2. I have added the technical methods. 

3. I have added the corresponding references.

---

## [Editor Report · Decision Letter 1]

1 Aug 2023

PONE-D-23-03632R1Propolis alleviates ulcerative colitis injury by inhibiting the protein kinase C - transient receptor potential cation channel subfamily V member 1 - calcitonin gene-related peptide/substance P (PKC-TRPV1-CGRP/SP) signaling axisPLOS ONE

Dear Dr. Jing,

Thank you for submitting your manuscript to PLOS ONE. After careful consideration, we feel that it has merit but does not fully meet PLOS ONE’s publication criteria as it currently stands. Therefore, we invite you to submit a revised version of the manuscript that addresses the points raised during the review process.

We look forward to receiving your revised manuscript.

Kind regards,

Suprabhat Mukherjee, Ph.D.

Academic Editor

PLOS ONE

Journal Requirements:

Additional Editor Comments:

Authors have not addressed my (Academic Editor) comments .

---

## [Author Response · Author response to Decision Letter 1]

14 Aug 2023

Dear editor,

 Thanks for your opinions on the revision of the paper. I have checked the references to make sure they are complete and correct, and the additions and modifications have been highlighted in ‘Revised Manuscript with Track Changes’.

---

## [Editor Report · Decision Letter 2]

22 Aug 2023

PONE-D-23-03632R2Propolis alleviates ulcerative colitis injury by inhibiting the protein kinase C - transient receptor potential cation channel subfamily V member 1 - calcitonin gene-related peptide/substance P (PKC-TRPV1-CGRP/SP) signaling axisPLOS ONE

Dear Dr. Jing,

Thank you for submitting your manuscript to PLOS ONE. After careful consideration, we feel that it has merit but does not fully meet PLOS ONE’s publication criteria as it currently stands. Therefore, we invite you to submit a revised version of the manuscript that addresses the points raised during the review process.

ACADEMIC EDITOR: Please address my comments given during the first decision.

We look forward to receiving your revised manuscript.

Kind regards,

Suprabhat Mukherjee, Ph.D.

Academic Editor

PLOS ONE

Journal Requirements:

Additional Editor Comments:

Authors need to address the comments (provided during first decision) of the academic editor.

---

## [Author Response · Author response to Decision Letter 2]

17 Sep 2023

1. I am ensure that my manuscript meets PLOS ONE's style requirements, including the file naming.

2. I have provided additional information regarding the experiments involving animals in Methods section, and included details on (1) methods of sacrifice, (2) methods of anesthesia and/or analgesia, and (3) efforts to alleviate suffering.

3. I have included a title page within my main document.

4. I have stated what role the funders took in the study. 

5. I have provided ORCID iD for the corresponding author.

6. Introduction has been amended and it contains the following: (1)Pharmacology of Propolis. (2)Mention the typical aetiology of UC and link PKC-TRPV1-CGRP/SP signaling pathway to that.

7. The follow latest citations have been cited and added:

(1) https://doi.org/10.1016/j.mex.2023.102158

(2) https://doi.org/10.1016/j.jnutbio.2018.07.010

(3) https://doi.org/10.3390/vaccines11030525

8. The subtitles had be revised, “Stool monitoring” had be changed to “ Stool occult blood test ”, “Effect of propolis on the expression of TRPV1, PKC, CGRP, and SP in UC-derived colonic tissues” had be changed to “Expression of TRPV1, PKC, CGRP, and SP protein in colonic tissues”.

9. I have reviewed the corresponding references to ensure that it is complete and correct.

---

## [Editor Report · Decision Letter 3]

19 Sep 2023

PONE-D-23-03632R3Propolis alleviates ulcerative colitis injury by inhibiting the protein kinase C - transient receptor potential cation channel subfamily V member 1 - calcitonin gene-related peptide/substance P (PKC-TRPV1-CGRP/SP) signaling axisPLOS ONE

Dear Dr. Jing,

Thank you for submitting your manuscript to PLOS ONE. After careful consideration, we feel that it has merit but does not fully meet PLOS ONE’s publication criteria as it currently stands. Therefore, we invite you to submit a revised version of the manuscript that addresses the points raised during the review process.

We look forward to receiving your revised manuscript.

Kind regards,

Suprabhat Mukherjee, Ph.D.

Academic Editor

PLOS ONE

Journal Requirements:

Additional Editor Comments:

I can not see the suggested citations in the re-revised manuscript.

---

## [Author Response · Author response to Decision Letter 3]

25 Sep 2023

1. I am ensure that my manuscript meets PLOS ONE's style requirements, including the file naming.

2. I have provided additional information regarding the experiments involving animals in Methods section, and included details on (1) methods of sacrifice, (2) methods of anesthesia and/or analgesia, and (3) efforts to alleviate suffering.

3. I have included a title page within my main document.

4. I have stated what role the funders took in the study. 

5. I have provided ORCID iD for the corresponding author.

6. Introduction has been amended and it contains the following: (1)Pharmacology of Propolis. (2)Mention the typical aetiology of UC and link PKC-TRPV1-CGRP/SP signaling pathway to that.

7. The follow latest citations have been cited and added:

(1) https://doi.org/10.1016/j.mex.2023.102158 (reference 5)

(2) https://doi.org/10.1016/j.jnutbio.2018.07.010 (reference 21)

(3) https://doi.org/10.3390/vaccines11030525 (reference 22)

8. The subtitles had be revised, “Stool monitoring” had be changed to “ Stool occult blood test ”, “Effect of propolis on the expression of TRPV1, PKC, CGRP, and SP in UC-derived colonic tissues” had be changed to “Expression of TRPV1, PKC, CGRP, and SP protein in colonic tissues”.

9. I have reviewed the corresponding references to ensure that it is complete and correct.

---

## [Editor Report · Decision Letter 4]

27 Oct 2023

Propolis alleviates ulcerative colitis injury by inhibiting the protein kinase C - transient receptor potential cation channel subfamily V member 1 - calcitonin gene-related peptide/substance P (PKC-TRPV1-CGRP/SP) signaling axis

PONE-D-23-03632R4

Dear Dr. Jing,

We’re pleased to inform you that your manuscript has been judged scientifically suitable for publication and will be formally accepted for publication once it meets all outstanding technical requirements.

Kind regards,

Suprabhat Mukherjee, Ph.D.

Academic Editor

PLOS ONE
---

## [Editor Report · Acceptance letter]

28 Dec 2023

PONE-D-23-03632R4 

PLOS ONE

Dear Dr. Jing, 

I'm pleased to inform you that your manuscript has been deemed suitable for publication in PLOS ONE. Congratulations! Your manuscript is now being handed over to our production team.

Kind regards, 

on behalf of

Dr. Suprabhat Mukherjee 

Academic Editor

PLOS ONE